

# The unintended consequence of $SO_2$ and $NO_2$ regulations over China: increase of ammonia levels and impact on $PM_{2.5}$ concentrations

Mathieu Lachatre[1], Audrey Fortems-Cheiney[1], Gilles Foret[1], Guillaume Siour[1], Gaëlle Dufour[1], Lieven Clarisse[3], Cathy Clerbaux[2,3], Pierre-François Coheur[3], Martin Van Damme[3], and Matthias Beekmann[1]

[1]Laboratoire Inter-Universitaire des Systèmes Atmosphériques (LISA), UMR CNRS 7583, Université Paris Est Créteil et Université Paris Diderot, Institut Pierre Simon Laplace, Créteil, France.
[2]LATMOS/IPSL, Sorbonne Université, UVSQ, CNRS, Paris, France.
[3]Université libre de Bruxelles (ULB), Spectroscopie Atmosphérique, Service de Chimie Quantique et Photophysique, Brussels, Belgium.

**Correspondence:** Mathieu Lachatre (Mathieu.lachatre@lisa.u-pec.fr)

**Abstract.** Air pollution, reaching hazardous levels in many Chinese cities has been a major concern in China over the past decades. New policies have been applied to regulate anthropogenic pollutant emissions, leading to changes in atmospheric composition and in particulate matter (PM) production. Increasing levels of atmospheric ammonia columns have been observed by satellite during the last years, in particular IASI observations reveal an increase of these columns by 15 % and 65 % from 2011 to 2013 and 2015, respectively, over Eastern China. In this paper we have performed model simulations for 2011, 2013 and 2015 in order to understand the origin of this increase, and in particular to quantify the link between ammonia and the inorganic components of particles: $NH_{4(p)}^+$ / $SO_{4(p)}^{2-}$ / $NO_{3(p)}^-$. Interannual change of meteorology can be excluded as a reason: year 2015 meteorology leads to enhanced sulphate production over Eastern China which increases the ammonium and decreases the ammonia content which is contrary to satellite observations. Reductions in $SO_2$ and $NO_X$ emission between 2011 and 2015 of respectively -37.5 and -21 %, as constrained from satellite data, lead to decreased inorganic matter (by 14 % for $NH_{4(p)}^+$ + $SO_{4(p)}^{2-}$ + $NO_{3(p)}^-$). This in turn leads to increased gaseous $NH_{3(g)}$ tropospheric columns, by as much as 24 % and 49 % (sampled corresponding to IASI data availability) from 2011 to 2013 and 2015 respectively, and thus can explain most of the observed increase.

## 1 Introduction

Particulate matter (PM) pollution poses serious health concerns all over the world, and particularly over China (Cohen et al., 2017; Landrigan et al., 2017). Among PM precursors, several studies (Seinfeld and Pandis, 2006; Schaap et al., 2004; Lelieveld et al., 2015; Bauer et al., 2016; Pozzer et al., 2017) pointed out the importance of ammonia ($NH_{3(g)}$), whose main source is agriculture, and which acts as a limiting species in the formation of fine particulate matter ($PM_{2.5}$, particulate matter with an aerodynamic diameter less than 2.5 μm, Banzhaf et al., 2013). Balance between $SO_2$, $NO_X$ and $NH_3$ emissions will define



cation or anion limited regimes of inorganic particulate matter formation, and this is key for PM control policies (Paulot et al., 2016; Fu et al., 2017). Yet, an increase in the ammonia atmospheric content over China has been observed by the satellite instrument AIRS, with a trend of $+2.27\,\%.\mathrm{yr}^{-1}$ from 2002 to 2016 (Warner et al., 2017). IASI satellite observations (Van Damme et al., 2017) also show an increase expense of $NH_{3(g)}$ column over China between 2011 and 2015, with a sharper trend

between 2013 and 2015. Several factors could explain this enhancement: it could be an increase of $NH_{3(g)}$ Chinese emissions, due to agricultural activities representing 80-90 % of total ammonia emissions in China (Kang et al., 2016). However, ammonia emissions appear to have reached a maximum in 2005 and been almost constant since then (Kang et al., 2016). Another study (Zhang et al., 2018) estimates a 7 % increase of $NH_3$ emissions between 2011 and 2015, much lower than the observed trends. Additional sources, such as biomass burning and associated ammonia emissions, do not show particular trends (Wu et al.,

2018). Consequently, the increase of atmospheric $NH_{3(g)}$ concentrations over China does not seem to be explained by changes in ammonia emissions.

Likewise, the rise of ammonia concentrations over China could be explained by increased $NH_{3(g)}$ exhaled from inorganic PM due to a rise in temperature, as shown by Riddick et al. (2016), meteorological variations would change both the $NH_{3(g)}$ volatilization and the equilibrium between ammonia, ammonium nitrate $NH_4NO_3$ and nitric acid $HNO_3$.

Finally, a decrease in sulphate and total nitrate availability caused by $SO_X$ ($SO_2 + SO_{4(p)}^{2-}$) and $NO_X$ ($NO + NO_2$,) emission reductions (Liu et al., 2017; de Foy et al., 2016) could leave more ammonia in the gas phase, since less ammonium is required to neutralize particle-phase acids, following a mechanism already observed by Schiferl et al. (2016) over the United States. Such a decrease in $SO_2$ emissions also occurred over China since 2011 (Koukouli et al., 2018). Its impact on atmospheric ammonia concentrations has not been quantified yet. Wang et al. (2013) examined the change of Chinese sulphate-nitrate-

ammonium aerosols due to anthropogenic emission changes of $SO_2$ and $NO_X$ from 2000 to 2015. However, they assumed an augmentation of $NO_X$ emissions from 2006 to 2015, and not the large decrease currently observed. A very recent study by (Liu et al., 2018, in discussion) suggests that ammonia increase comes from $SO_2$ emission policies, but does not take into account $NO_X$ emissions, nor a comparison to observed $NH_3$ observations.

Our goal here is to understand the factors controlling atmospheric ammonia concentrations over China for recent years. In

order to identify the drivers of $NH_{3(g)}$ variability over China, we ran different sensitivity studies with a regional chemistry-transport model, isolating the impacts of i) meteorological conditions and ii) decrease of anthropogenic $SO_2$ and $NO_X$ emissions. Section 2 presents the regional chemistry-transport model (CTM) CHIMERE used in this study and the different settings of the performed sensitivity tests. It also presents IASI $NH_3$ column and surface PM observations. Section 3 gives the results of the sensitivity tests and shows the impacts of meteorological conditions and of emission changes on ammonia and ammo-

nium concentrations. The simulations are also evaluated by comparison of the modelled $PM_{2.5}$ concentrations with surface measurements. Finally, the modelled $NH_{3(g)}$ column inter-annual variability is compared to the one retrieved from satellite (IASI) data.





## 2 Material and method

### 2.1 IASI Satellite observations

$NH_{3(g)}$ column observations from space are provided by the IASI instrument (Infrared Atmospheric Sounding Interferometer, operating between 3.7 - 15.5 µm) on board the Metop-A European satellite (Clarisse et al., 2009; Van Damme et al., 2018). The
algorithm used to retrieve $NH_3$ columns from the radiance spectra is described in Whitburn et al. (2016) and Van Damme et al. (2017). For this study we used the 'climate' dataset ANNI-NH3-v2.2R-I, relying on ERA-Interim ECMWF meteorological input data rather than the operationally provided Eumetsat IASI Level 2 (L2) data used for the standard near-real-time version, that is coherent in time (excepted for the cloud coverage flag) and can therefore be used to study interannual $NH_{3(g)}$ variability over East China between 2011 and 2015 (Figure 1). We used the daily satellite information from morning orbits (at ∼9.30
am LT), to have daily information on IASI data availability. The IASI total columns are first averaged into daily "super-observations" (average of all individual IASI data within the $0.25° \times 0.25°$ resolution of CHIMERE). The annual gridded means of Figure 1 are calculated from these gridded daily "super-observations". In this study, and as suggested in Van Damme et al. (2017), we do not apply a selection to the IASI observations. In 2011, a mean value of $4.7 \times 10^{15}$ molecules.cm$^{-2}$ is observed for East China, increasing to $5.36 \times 10^{15}$ molecules.cm$^{-2}$ in 2013 (+15 %), and $7.76 \times 10^{15}$ molecules.cm$^{-2}$ in 2015
(+65 %).

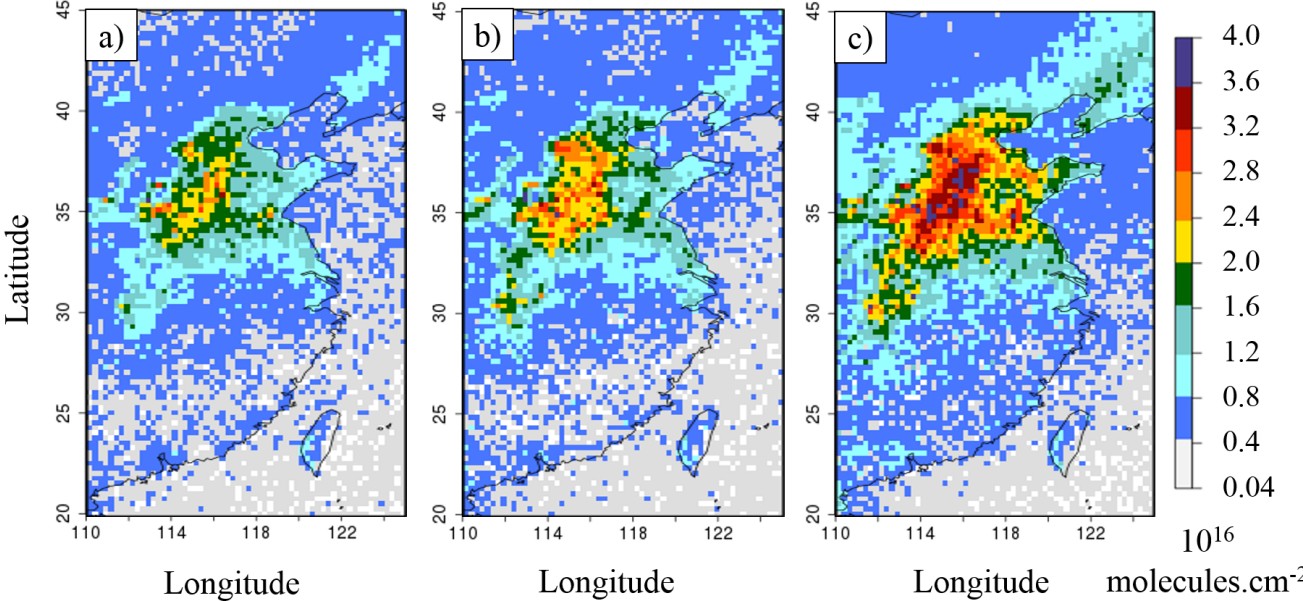

**Figure 1.** IASI instrument $NH_{3(g)}$ total columns over East China (110°E - 125°E; 20°N - 45°N), in $10^{16}$ molecules.cm$^{-2}$ for a) 2011, b) 2013 and c) 2015. In this figure, the available observations have been averaged on a $0.25° \times 0.25°$ grid.




## 2.2 The chemistry-transport CHIMERE model and updated $NO_X$ and $SO_2$ emissions

CHIMERE (2014b version) is a 3D chemistry-transport regional model (Menut et al., 2013; Mailler et al., 2017, www.lmd.polytechnique.fr/chimere/, last consulted 09/15/18) run here over a $0.25° \times 0.25°$ regular grid, on a domain integrally covering China's territory (72°30'E - 145°E; 17°30'N - 55°N). The domain includes 290 (longitude) $\times$ 150 (latitude) grid cells and

17 vertical layers, with altitude going from the ground to 200 hPa (about 12 km), with 8 layers within the first two kilometers. Meteorological fields are provided by ECMWF meteorological forecasts (Owens and Hewson, 2018). To prescribe atmospheric boundaries and initial composition, climatological values are used from LMDZ-INCA global model (Szopa et al., 2008).

Biogenic emissions are calculated taking into account meteorological parameters with the MEGAN-v2 model (Guenther et al., 2006). We use the EDGAR-HTAP-v2.2 inventory delivered for the year 2010 (Janssens-Maenhout et al., 2015), to

prescribe the anthropogenic emissions for the simulation. Chinese emissions in EDGAR-HTAP-v2.2 are derived from the MEIC inventory developed by Tsinghua University, the $NH_{3(g)}$ emission inventory from Peking University (Huang et al., 2012), and the REAS inventory (Kurokawa et al., 2013) to fill the remaining gaps. The respective total annual emissions of $SO_2$, NO and $NH_{3(g)}$ for 2010 are 42.4 Mt, 25.2 Mt, and 20.3 Mt (Table 1) and the spatial distributions of these emissions are represented in Figure 2.

**Table 1.** Annual budgets of the EDGAR-HTAP-v2.2 inventory and of emissions corrected from the OMI instrument (this work), for $SO_2$, NO and $NO_2$, over Asia and over East China, in Mt. The "Asian" domain corresponds to our full domain (72°30'E - 145°E; 17°30'N - 55°N), and the "East China" domain corresponds to a smaller domain (110°E - 125°E; 20°N - 45°N), displayed by a dark rectangle in Figure 2.

| **Domains** | Species | Reference Emissions EDGAR-HTAP-v2.2 for 2010 | Emissions derived for 2013 | Emissions derived for 2015 |
|---|---|---|---|---|
| **Asia** | $NH_3$ | 20.3 Mt | = | = |
| | $SO_2$ | 42.4 Mt | +3.8 Mt / +08.9 % | -6.8 Mt / -16.0 % |
| | NO | 25.2 Mt | +1.2 Mt / +04.8 % | -2.2 Mt / -08.7 % |
| | $NO_2$ | 4.1 Mt | +0.2 Mt / +04.9 % | -0.2 Mt / -04.9 % |
| **East China** | $NH_3$ | 6.0 Mt | = | = |
| | $SO_2$ | 19.7 Mt | -.4.7 Mt / -23.8% | -7.4 Mt / -37.5 % |
| | NO | 13.0 Mt | < +0.1 Mt / < +1.0% | -2.8 Mt / -21.5 % |
| | $NO_2$ | 1.85 Mt | < +0.1 Mt / +2.5 % | -0.4 Mt / -21.0 % |

We used the most recent HTAP emission inventory built for the year 2010 to simulate emissions of the year 2011, assuming that both years have similar emissions. Initially, for 2011, OMI and CHIMERE $NO_2$ columns have been compared, we obtained a daily variation of the Pearson correlation coefficient of 0.78 and an annual mean bias of -7 %. To understand the impact of the $NO_X$ and $SO_2$ emissions reduction observed over China in 2013 and 2015, emissions need to be updated and the observed





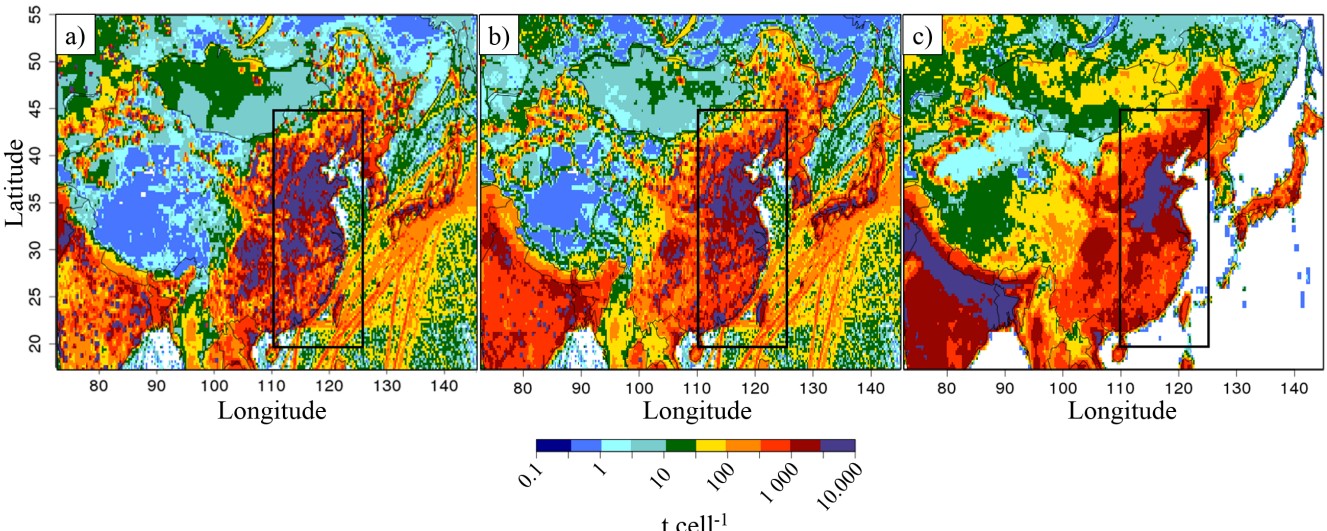

**Figure 2.** EDGAR-HTAP-v2.2 emissions for the year 2010 for a) $SO_2$, b) NO and c) $NH_3$. Units are in t.cell$^{-1}$ (cell size of $0.25° \times 0.25°$). The black rectangle represents the so-called East China domain.

variability of satellite columns has been used to update emissions, as in Palmer et al. (2006). Here NO, $NO_2$ and $SO_2$ emissions have been updated following the variability observed by OMI between 2011, 2013 and 2015. We assume that $NO_2$ variation are piloted by $NO_X$ emissions changes. The yearly gridded relative variabilities seen by the OMI instrument between 2011 and 2013, and between 2011 and 2015, are applied to daily prior anthropogenic $SO_2$, NO and $NO_2$ emissions. $H_2SO_4$ emissions

represent only a small fraction of SOx (1 %) and, consequently, are not updated here. Equation 1 has been applied for each pixel of our regional grid for 2015 and 2013, where $i$ can be $SO_2$, $NO_2$ or NO and $j$ the pixel number:

$$Emis_{(year,i,j)} = Emis_{(2010,i,j)} \times \left( \frac{Col_{(year,i,j)}}{Col_{(2011,i,j)}} \right) \tag{1}$$

$Emis_{(2010)}$ represents the "reference" HTAP emission inventory, $Col_{(2011)}$ and $Col_{(year)}$ the respective OMI satellite observation values for 2011 and 2015 (or 2013). Correction for the emission inventory are shown on Figure 3. Values of derived

emissions have been limited to 500 % of initial values, to avoid generating some outliers values located in North Korea.

It should be noted that our annual average based update does not modify the seasonal cycle for $NO_X$ and $SO_2$ (with a maximum in winter, see Figure S1 in supplement file). In the model, spatial distributions for $NO_X$ and $SO_2$ emissions appear to have similar structure (see Figure 2). Ammonia emissions present a significant seasonal cycle with emissions higher during summer than during the other seasons. On a molar basis, $NH_{3(g)}$ emissions are low compared to $SO_2$ and $NO_X$ emissions,

except in summer when $NH_{3(g)}$ emissions are in excess compared to $SO_2$ alone (Figure S1 in supplement file).

When applying Equation 1, the temporal evolution of OMI $NO_2$ columns leads to a decrease in $NO_X$ emissions (Figure S3 in supplement file and Figure S4 in supplement file), particularly after 2013 (+1 % in 2013 and -21 % in 2015 as compared to



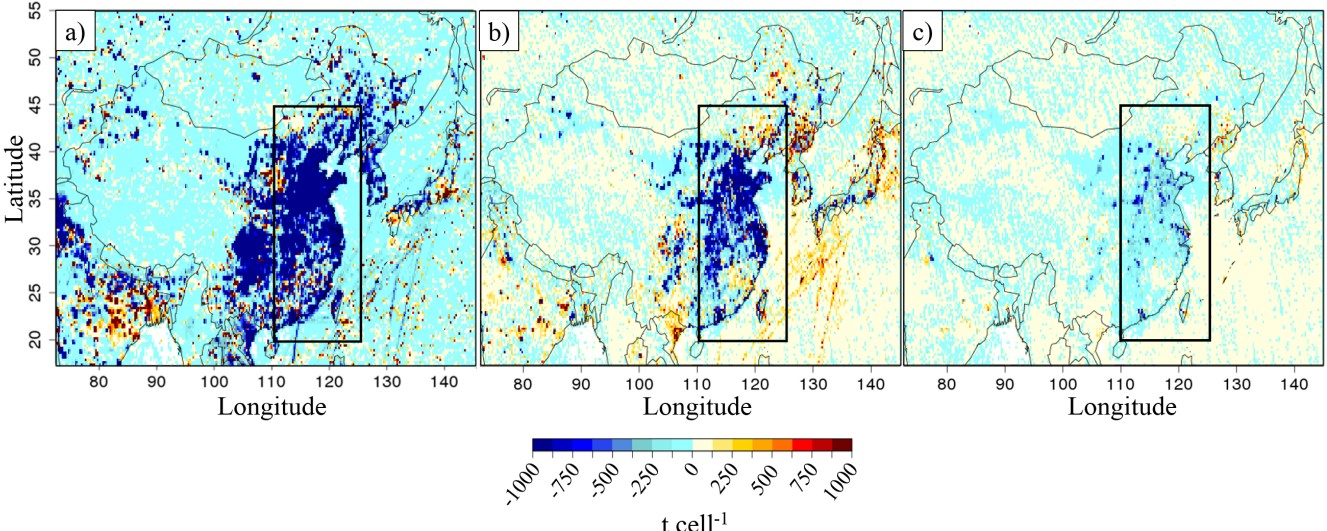

**Figure 3.** Annual differences between the EDGAR-HTAP-v2.2 inventories for the year 2010 and the emissions corrected from OMI for the year 2015, for a) $SO_2$, b) NO and c) $NO_2$. Units are in t.cell$^{-1}$ (cell size of $0.25° \times 0.25°$).

2011). Liu et al. (2017) derived similar $NO_X$ emission changes with the exponentially modified Gaussian method of Beirle et al. (2011) (e.g., decrease of 21 % of $NO_X$ emissions within Chinese cities between 2011 and 2015). We have also compared our updated inventories for $NO_X$ and $SO_2$ with the DECSO v5 inventories calculated with an inverse modelling method based on an extended Kalman Filter (Mijling and Zhang, 2013; Ding et al., 2017, www.globemission.eu/, last consulted 07/04/2018).

For DECSO, the a priori $NO_X$ anthropogenic emissions are taken from the EDGAR-v4.2 inventory whereas our prior emissions come from EDGAR-HTAP-v2.2. EDGAR-v4.2 does not take into account shipping emissions as shown on Figure S2 in supplement file. Our annual evolutions are consistent with DECSO trend estimates (+1 % in 2013 and -14 % in 2015, compared to DECSO in 2011).

The OMI $SO_2$ trends imply (following Equation 1) a continuous decrease of $SO_2$ emissions from 2011 to 2015 (-24 %

in 2013 and -37 % in 2015 compared to 2011, see Table 1 and Figure 2a). This decreasing trend is consistent with trend estimates of the $SO_2$ DECSO, -11 % in 2013 and the -25 % in 2014 compared to 2011. Nevertheless, our total $SO_2$ emissions in 2013 seem to be underestimated compared to the DECSO annual estimates, (-15 % in 2011, -25 % in 2013). It should be noted here that our method to update emissions has some limitations. Indeed all the variability of satellite tropospheric columns is attributed to emissions, without taking into account variability associated to meteorology, transport, chemistry or

instrumental degradation. However, the comparison with independent emission estimations shows a rather good consistency. We then consider that our estimated emission inventories are realistic enough to conduct sensitivity tests.



## 2.3 ISORROPIA

The composition and phase state of inorganic aerosol in thermodynamic equilibrium with gas phase precursors are calculated using the ISORROPIA V2006 module (Nenes et al., 1998). The CHIMERE CTM calculates the thermodynamical equilibrium of the system: sulphate-nitrate-ammonium-water for a given temperature (in a range [260 K-312 K]; increment: $\Delta K = 2.5$ K) and relative humidity (RH, in a range [0.3-0.99]; increment: $\Delta RH = 0.05$). Considering temperature, relative humidity, TN (TN = $NO_{3(p)}^-$ + $HNO_3$; Total nitrates), TA (TA = $NH_{4(p)}^+$ + $NH_{3(g)}$; Total ammonia) and TS (TS = $SO_{4(p)}^{2-}$ + $H_2SO_4$; Total Sulphates) the gas to particle partitioning of $NH_{4(p)}^+$ / $NH_{3(g)}$ and $HNO_{3(g)}$ / $NO_{3(p)}^-$ are calculated using tabulated values. Depending on equilibrium calculation, both absorption and desorption of ammonia are represented in model. Within CHIMERE, a kinetic approach is also added to simulate transport barriers for the gas to the particle phase and vice versa. Over Europe, a recent study, conducted with CHIMERE and the ISORROPIA V2006 module, showed that higher temperature and lower RH will promote $NH_{3(g)}$ at the expense of $NH_{4(p)}^+$ (Petetin et al., 2016).

## 2.4 Set-up of the performed sensitivity tests

Six experiments have been performed to discriminate factors that control the ammonia atmospheric budget over China during the recent years. Configurations corresponding to the six experiments are described in Table 2.

The "2011" reference simulation serves as a baseline for comparison with the other simulations (under different meteorological and emissions scenarios). The "A" labelled simulations are used to quantify influence of meteorological parameters, the "B" labelled simulation to quantify influence of $SO_2$ emissions variations, and "C" labelled simulations to quantify the influence of both $SO_2$ and $NO_X$ emissions variations. The 2013A and 2015A simulations were performed keeping emissions at 2010 levels, but the meteorology was updated, to isolate effects of meteorological variability on simulated ammonia levels. Among scenarios performed in our study, the 2015B simulation uses the same meteorology as 2015A but with 2015 $SO_2$ emission giving information on the role of Chinese $SO_2$ emissions reduction on the $NH_{3(g)}$ atmospheric content. Finally, the simulation 2015C uses the same meteorology than the 2015A one, but using both $SO_2$ and $NO_X$ emissions corrected with 2015 OMI observations (see Table 2).

As the 2013 updated $NO_X$ emissions are similar to the initial EDHAR-HTAP-v2.2 ones used for the 2011 reference simulation, we expect similar results with the 2013B and 2013C simulations. Consequently, the 2013B simulation has not been conducted. 2013C and 2015C are then the key simulations of our study showing the combined effect of $SO_2$ and $NO_X$ emission reductions and of meteorology on atmospheric ammonia.

## 3 Results and Discussion

### 3.1 Impact of meteorological conditions on the ammonia/ammonium/sulphate/nitrate system

Three scenario simulations using the same emissions but different meteorological conditions (2011A, 2013A and 2015A) have been performed to understand the role of meteorology for ammonia atmospheric concentrations. As our study uses satellite



**Table 2.** Description of the different experiments performed by the regional chemistry-transport model CHIMERE.

| **Name** | Meteorology | SO$_2$ emissions | NO$_X$ emissions | Objectives of the simulation |
|---|---|---|---|---|
| **2011A** | ECMWF 2011 | EDGAR-HTAP-v2.2 2010 | EDGAR-HTAP-v2.2 2010 | Baseline simulation |
| **2013A** | ECMWF 2013 | EDGAR-HTAP-v2.2 2010 | EDGAR-HTAP-v2.2 2010s | Sensitivity to meteorology |
| **2013C** | ECMWF 2013 | Deduced from OMI for 2013 | Deduced from OMI for 2013 | Sensitivity to SO$_2$ and NO$_X$ emission reduction |
| **2015A** | ECMWF 2015 | EDGAR-HTAP-v2.2 2010 | EDGAR-HTAP-v2.2 2010 | Sensitivity to meteorology |
| **2015B** | ECMWF 2015 | Deduced from OMI for 2015 | EDGAR-HTAP-v2.2 2010s | Sensitivity to SO$_2$ emission reduction |
| **2015C** | ECMWF 20115 | Deduced from OMI for 2015 | Deduced from OMI for 2015 | Sensitivity to SO$_2$ and NO$_X$ emission reduction |

observations giving tropospheric trace gas columns for different purposes, also model results will be generally presented as vertical columns (reaching from ground to 12 km height). However, most of the column content can generally be found within first 2.5 km, close to the ground (more than 90 % of NH$_{3(g)}$ is located within first 2.5 km in CHIMERE). Hence, to study the meteorological influence on ammonia, we restrict the comparison to 0 to 2.5 km (which corresponds to about 720 hPa) partial columns, as we want to average for meteorological parameters for a height, which should be representative for conditions where pollutants are located.

Figure 4 shows NH$_{3(g)}$ columns in 2011 (Figure 4a), and simulated variations depending on meteorological conditions for 2013 (Figure 4 b) and 2015 (Figure 4c). It can be observed that the so-called "East China" area includes regions with the highest NH$_{3(g)}$ values (except India Gangetic valley). Over the East China area, meteorological conditions affect NH$_{3(g)}$ columns: an increase of 4 % is simulated in 2013 whereas a decrease of 7 % is simulated in 2015. In addition, it can be observed that over the south China area (see black rectangle in Figure 4 b-c) that ammonia decreases for 2013 and 2015. Figure 5 shows that ammonia changes caused by meteorological variations are opposite to sulphate and ammonium changes, in both cases, for 2013 and 2015. Indeed, meteorological conditions in 2015 (displayed on Figure S5 in supplement file) promoted the formation of ammonium and sulphates (+6 % and +12 % respectively) in 2015A compared to 2011A, and a decrease of nitrates (-13 %; Figure 5). For 2013, changes of NH$_{3(g)}$ are opposite to ammonium (-7 %) and sulphates (-7 %), with a slight increase of NH$_{3(g)}$ (+4 %). It appears that changes in the NH$_{3(g)}$/ NH$_{4(p)}^+$ ratio are correlated with SO$_{4(p)}^{2-}$ changes. This can be explained from





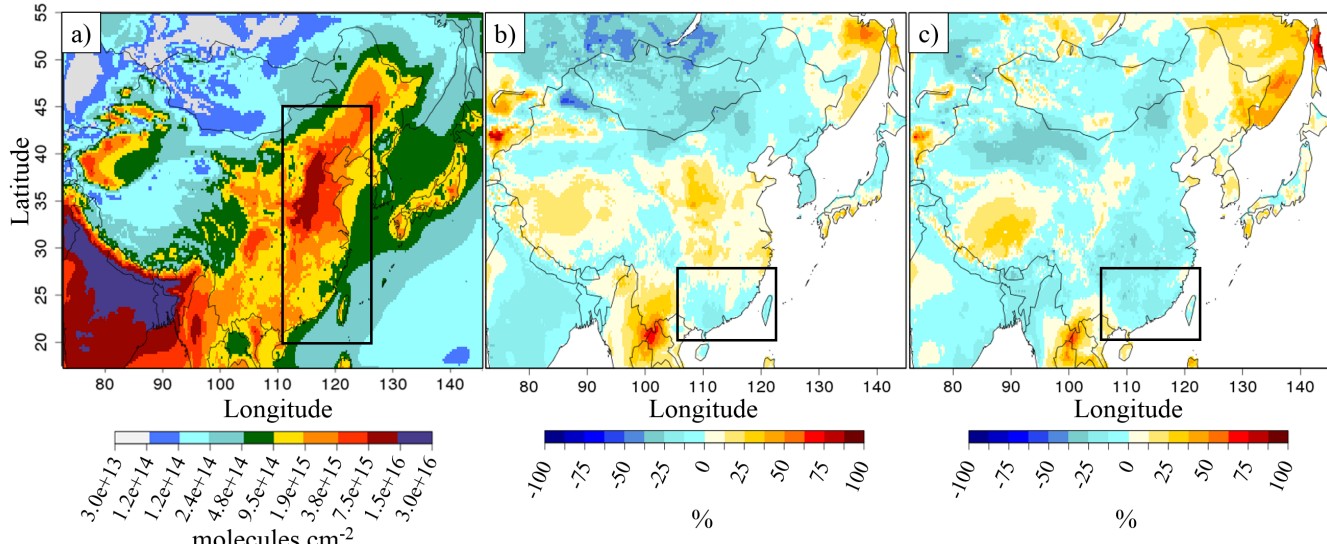

**Figure 4.** a) Ammonia columns (molec.cm$^{-2}$) for the simulation 2011A; b) relative differences (%) of ammonia columns between 2013A and 2011A; c) relative differences between 2015A and 2011A. Note that as values over the sea mostly represent only small variations, we do not show them here and in the following figures.

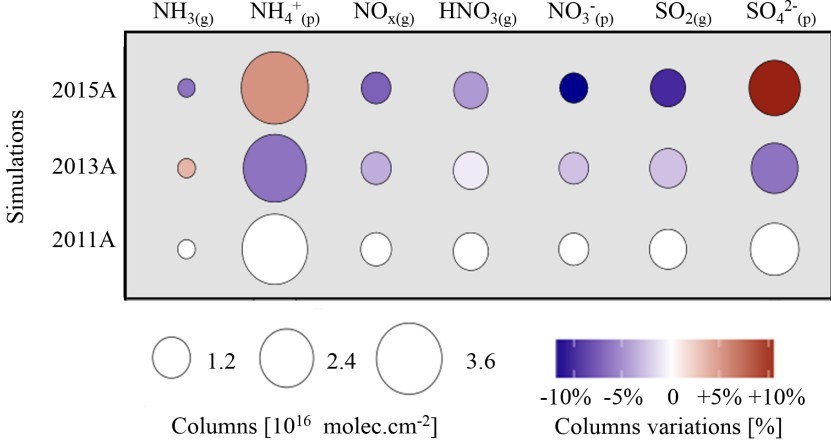

**Figure 5.** Gaseous and particulate inorganic species variation for the East China region. Disk surfaces are proportional to columns amount (molec.cm$^{-2}$) and colors indicate relative evolutions compared to 2011A (%).

well-known neutralisation reactions in the gas (Reaction R1) or aqueous phase (Reaction R2 and R3):

$$2NH_{3(g)} + H_2SO_{4(g)} \rightleftharpoons (NH_4)_2SO_{4(p)} \tag{R1}$$

$$NH_{3(aq)} + H_2SO_{4(aq)} \rightleftharpoons NH_{4(aq)}^+ + HSO_{4(aq)}^- \tag{R2}$$

$$NH_{3(aq)} + HSO_{4(aq)}^- \rightleftharpoons NH_{4(aq)}^+ + SO_{4(aq)}^{2-} \tag{R3}$$




$SO_2$ oxidation to $SO_{4(p)}^{2-}$ can happen in gas phase, from process Reaction R5, in which OH is the oxidant, produced from Reaction R4 reaction that is linked to humidity:

$$O(^1D) + H_2O_{(g)} \rightarrow 2OH \tag{R4}$$

$$OH + S^{(+IV)}O_{2(g)} \rightarrow H_2S^{(+VI)}O_{4(g)} \tag{R5}$$

Sulphate production can also occur in the aqueous phase, by $SO_{2(aq)}$ oxidation with $O_3$ or $H_2O_2$, yielding sulfuric acid which then can be neutralised to form ammonium sulphate (Hoyle et al., 2016). The extent of this reaction chain depends on cloud liquid water content. It is initiated by a solution of $SO_{2(g)}$ in the water phase ($SO_2.H_2O_{(aq)}$):

$$SO_{2(aq)} + H_2O_{(aq)} \rightleftharpoons SO_2.H_2O_{(aq)} \tag{R6}$$

$$SO_2.H_2O_{(aq)} + O_{3(aq)} \rightarrow HSO_{4(aq)}^+ + H^+ + O_2 \tag{R7}$$

While the aqueous phase pathway is globally dominant, the gas phase pathway can be also major under dry conditions (e.g., Seinfeld and Pandis, 2006). In our study, we did not investigate the relative importance of both pathways, because this would have required inclusion of specific tagging. Nevertheless, for both of them, RH increase favours $SO_{4(p)}^{2-}$ production, or through increased production of OH radicals in the gaseous phase (for a given temperature, so that also specific humidity increases), either through a larger cloud liquid water content (Hedegaard et al., 2008).

In 2013, annual mean temperature and annual mean RH have respectively increased by +0.7 K and decreased by -0.9 % compared to 2011, also cloud liquid water relative variation shows a decrease of -7 %. In 2015, temperature and annual mean RH have respectively increased by 1 K and increased by +1.3 % compared to 2011, and cloud liquid water relative variation presents an increase of +20 %. Accordingly, the total sulphate-nitrate-ammonium (called pSNA hereafter) production is promoted, +7 % in 2015A compared to 2011A, explaining thus the decrease in the $NH_{3(g)}$ columns by -7 %. On the contrary, meteorological conditions in 2013 (decrease of RH and liquid water) decreased the formation of pSNA (Figure 5). Consequently, its production was lower by about 6 % in 2013A than in 2011A, and $NH_{3(g)}$ columns larger by +4 %.

These relationships between meteorological parameters and $NH_{3(g)}$ columns can also be documented by correlation statistics (Figure 6). Indeed, an inverse correlation between monthly RH and $NH_{3(g)}$ column variations over the previously defined East China domain is shown in Figure 6a, with Pearson correlation coefficients of -0.47 and -0.56, in 2013 and 2015, respectively. An even more pronounced negative correlation is also observed on a daily basis, with correlation coefficients for 2013 and 2015 of -0.71 and -0.61 respectively. When RH increases, the production of $NH_{4(p)}^+$ from $NH_{3(g)}$ also increases. The largest difference between 2013A and 2015A is observed in November and December 2013 and 2015, when RH variations and $NH_{3(g)}$ column variations are opposite (Figure 6a). Figure 6b shows that temperature changes do not control the $NH_{3(g)}$ variation, as the Pearson correlation coefficients are -0.04 and 0.09, in 2013 and 2015. It also should be noted that decreases of ammonia and ammonium are observed over areas presenting an increase of rainfall frequencies (see Figure S6 in supplement file) in 2015 and 2013 (in the south of China, Guangxi, Guan-Dong provinces - see the black box in Figure 4). On the contrary, with rainfall frequencies lower than 90 days.yr$^{-1}$ (for rainfall above 1 mm.day$^{-1}$), and small changes in rainfall frequencies over North China Plain for 2011 and 2013, changes in wet deposition do not seem to impact significantly ammonia levels. Indeed,





low correlation is found between monthly rainfall frequencies variations and monthly ammonia variations over East China (Pearson correlation coefficients of -0.12 and -0.18 for 2013 and 2015 respectively). Finally it has been observed in this study

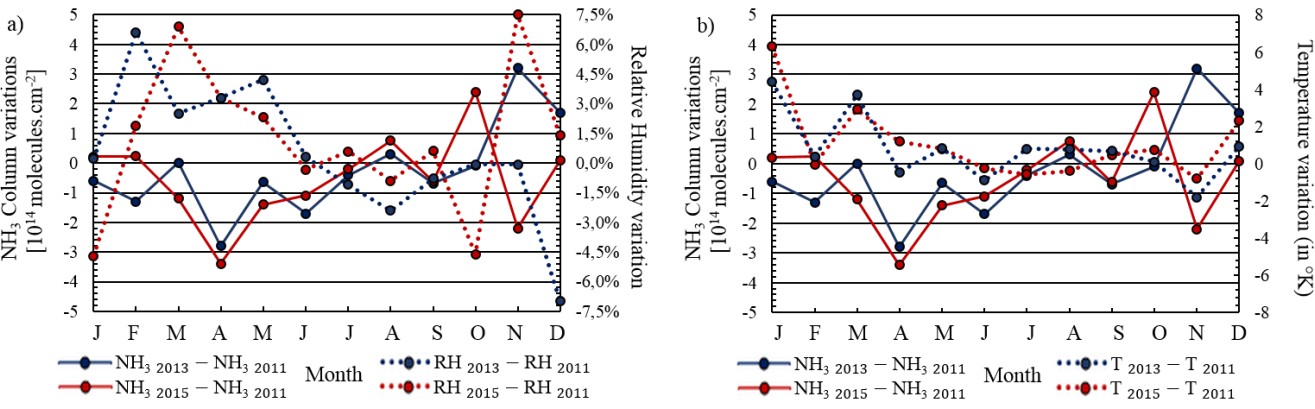

**Figure 6.** a) Monthly variation of $NH_{3(g)}$ partial columns and relative humidity in 2013 and 2015 compared to 2011, b) monthly variation of $NH_{3(g)}$ partial columns and temperature in 2013 and 2015 compared to 2011.

that total $NH_{3(g)} + NH_{4(p)}^+$ columns will vary depending on meteorological conditions. It should be noted that, if $NH_{3(g)}$ is favoured, as for 2015, the total content will decrease as $NH_{3(g)}$ lifetime is shorter than $NH_{4(p)}^+$ lifetime due to faster deposition.

## 3.2 Impact of $SO_2$ and $NO_X$ emission reduction on $NH_3$ columns and inorganic aerosol

### 3.2.1 Impact of $SO_2$ and $NO_X$ emission reduction on $NH_3$ columns

Figure 7b represents the impact of the $SO_2$ emission reduction on $NH_{3(g)}$ columns for the 2015B simulations. For year 2013, the comparison is made from the 2013C simulation (see Table 1, and Figure S7 in supplement file), since $NO_X$ emissions between 2011 and 2013 are similar. The $SO_2$ emission reduction (-24 % for 2013 and -37 % for 2015 as compared to 2011) strongly affects $NH_{3(g)}$ columns: they increase by +10 % over East China in the 2013C simulation compared to 2013A and by about +36 % in the 2015B simulation compared to 2015A. Thus, the effect of $SO_2$ reduction on $NH_{3(g)}$ columns appears to be non-linear, because $NH_{3(g)}$ interactions are not limited to $SO_2$. It should be noted that the column change is mainly controlled by changes between the surface and the first kilometres of altitude, as much of column content (>95 %) is located between the surface and 2.5 km of altitude, but as IASI and OMI satellite provide full column information, we present the CHIMERE entire column to be as consistent as possible with observations. In the two cases, the decrease of ammonia over western China and Mongolia (between 0 and -15 %, Figure 7b), where $NH_{3(g)}$ values are initially low (Figure 7a), remains small. Figure 7c shows the additional impact of $NO_X$ emission reductions, of about -21 % between 2015 and 2011, on the $NH_{3(g)}$ amount over East China (with the 2015C simulation, compared to 2015B). The additional increase of ammonia columns in the simulation 2015C, is about 15 % compared to the simulation 2015B ($SO_2$ emission decrease only) in the northern part of East China subdomain.





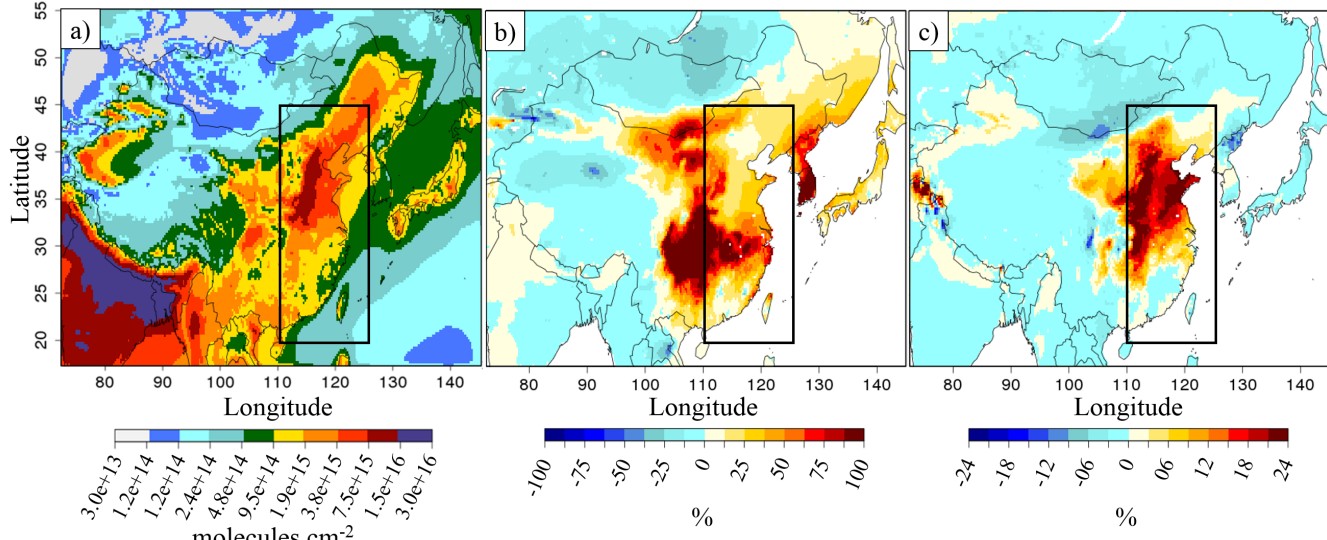

**Figure 7.** a) Ammonia columns in molec.cm$^{-2}$ for the simulation 2015A, b) relative differences of ammonia columns between the simulations from 2015B and 2015A, in % and c) additional relative differences of ammonia columns between the simulations from 2015C minus 2015B compared to 2015A, in %. The black rectangle is for East China domain.

Figure 8b represents the impact of the $SO_2$ emission reductions on $NH_{4(p)}^+$ columns with a decrease of about 14 % in the 2015B simulation compared to 2015A. The reduction of $NO_X$ emissions between 2015B and 2015C leads to an additional decrease of ammonium levels in the 2015C simulation, -4 % compared to 2015B for "East China" where the decrease is most pronounced Figure 8c. In addition, we have observed that ammonium columns have decreased by about 2 % over East China in the 2013C simulation compared to 2013A. The spatial anti-correlation observed between $NH_{4(p)}^+$ and $NH_{3(g)}$ is explained by less production of $NH_{4(p)}^+$ from the $NH_{3(g)}$ due to emission modifications. It is interesting to note that the $NO_X$ and $SO_2$ emission reduction impacts on $NH_{3(g)}$ and $NH_{4(p)}^+$ columns can differ depending on the areas (see Figure S8 in supplement file). In the 2015B simulation the ammonium production decreases most strongly in Sichuan province-Chongqing municipality (Black rectangle, Figure 8b), and there is a large scale decrease around $SO_2$ sources (Figure 2a). In the 2015C simulation, we can observe a larger decrease in North China region (red rectangle, Figure 8c), where ammonium nitrate is produced with freshly formed $HNO_{3(g)}$, following Reaction R8):

$$NH_{3(g)} + HNO_{3(g)} \quad \rightleftharpoons \quad NH_4NO_{3(p)} \tag{R8}$$

These relationships between sources and impacted regions are explained by the time needed for $SO_{4(p)}^{2-}$ and $NO_{3(p)}^-$ formation from $SO_2$ and $NO_X$ precursors, respectively several days and several hours to one day. Thus $SO_2$ to $H_2SO_4$ oxidation is more a large scale process (unless it happens in the aqueous phase), whereas nitrate formation proceeds closer to the sources. However, in case of lower atmospheric dispersion or high water content, as for example in Sichuan province-Chongqing municipality, which present particular geographic situation (i.e. located into an orographic depression), allowing sulphates to be formed close




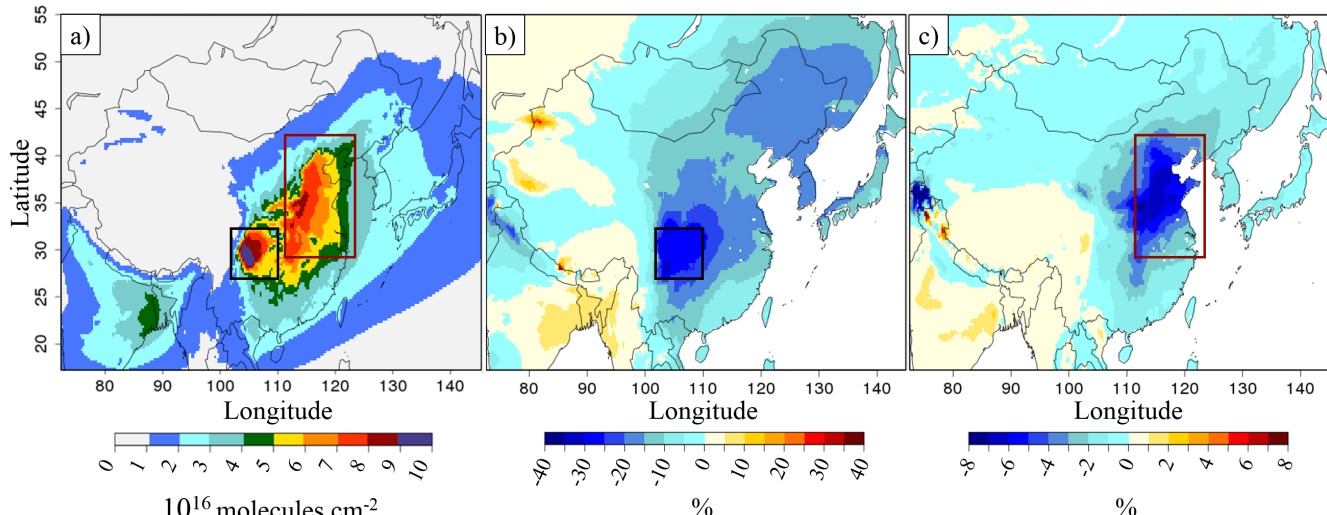

**Figure 8.** a) Ammonium columns in molec.cm$^{-2}$ for the simulation 2015A, b) relative differences of ammonium columns between the simulations from 2015B and 2015A, in % and c) additional relative differences of ammonium columns between the simulations from 2015C minus 2015B compared to 2015A, in %. The black rectangle is for the Sichuan province–Chongqing municipality, and the red rectangle is for North China.

to sources, and largely contributing to PM as high as 32 % in column (compared to 23 % over East China, $SO_{4(p)}^{2-}$ columns are displayed Figure S8 in supplement file).

### 3.2.2 Impact of $SO_2$ and $NO_X$ emission reduction on pSNA production

The emission update leads to changes in the pSNA production, as already suggested by changes in the $NH_{4(p)}^{+}$ columns. As
5 $SO_2$ columns are strongly decreased (-40 % for 2015B; -41 % for 2015C), less ammonium (-14 % for 2015B; -18 % for 2015C) is formed in the particulate phase from the reaction with sulphuric acid (Reaction R1, R2, R3), and more $NH_{3(g)}$ remains in the gas phase (+36 % for 2015B; +51 % for 2015C; see Figure 9 and Figure S9 in supplement file). These higher $NH_{3(g)}$ levels trigger a larger conversion (Reaction R8) of gaseous nitric acid into particulate $NO_{3(p)}^{-}$ (+33 %; Figure 9 and S10). In the 2015C simulation, the increase of $NO_{3(p)}^{-}$ is less notable (+11 %, because $NO_X$ emissions decrease), and ammonia columns show a
10 bigger increase than in 2015B over East China.

On the whole, the reduction of emissions in the 2015B and 2015C simulations leads to a reduction of the total pSNA PM production (e.g., -16.6 % and -18.5 %, respectively, compared to 2015A), mainly promoted by the reduction of $SO_2$ emissions. Among PM components, a decrease of the sulphate molar fraction is observed (from 32 % to 29 %) and the $NO_{3(p)}^{-}$ fraction increases, from 12 % to 15 % while $NH_{4(p)}^{+}$ molar fraction stays stable around 56 % (21 % of pSNA PM mass).




It can also be observed in these scenarios that for similar emissions and meteorology, TA=$[NH_{3(g)}]+[NH_{4(p)}^{+}]$ decreases in 2015B and slightly more in 2015C (Figure S9 in supplement file). The reasons are that ammonia is favoured compared to 2015A and that deposition is a more efficient process for ammonia.

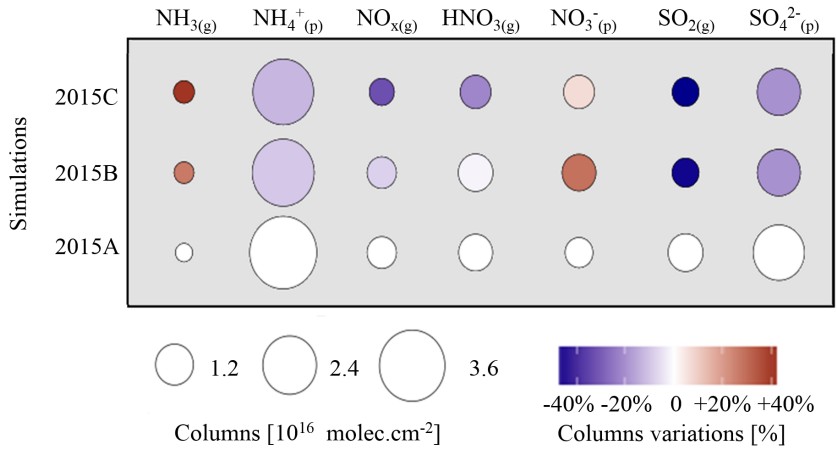

**Figure 9.** Gaseous and particulate inorganic species variation for the East China region. Disk surfaces are proportional to columns amount molec.cm$^{-2}$) and colours indicate relative evolutions compared to 2015A (%).

### 3.2.3 $NH_{3(g)}$: a key role for the regulation of PM pollution as a limiting reactant

The regime of nitrate production, limited either by $NH_{3(g)}$ or $HNO_{3(g)}$, can be evaluated using the $G_{ratio}$ calculation (Ansari and Pandis, 1999; Pinder et al., 2008, $G_{ratio}$ two-dimensionnal distribution is displayed in Figure S10 in supplement file). A negative value of the $G_{ratio}$ indicates that TA low availability is strongly limiting nitrate production in competition to sulphate production, while a value between [0-1] indicates a TA limitant situation, and a value greater than 1 indicates that TA is in excess. In our study, as we want to know if we are facing a cation or anion limited regime, we chose to adapt the $G_{ratio}$ to cations/anions ratio (C/A$_{ratio}$), easier to interpret with no negative values, as follows:

$$C/A_{ratio} = \frac{TA}{TN + 2 \times [SO_{4(p)}^{2-}]} \qquad (2)$$

where TA=$[NH_{3(g)}]+[NH_{4(p)}^{+}]$ is the total ammonia reservoir and TN=$[HNO_{3(g)}]+[NO_{3(p)}^{-}]$ is the total nitrate reservoir. When the C/A$_{ratio}$ is between [0-1] mol$_C$.mol$_A^{-1}$, we are in a cation limited regime, and when C/A$_{ratio}$ is larger than 1 mol$_C$.mol$_A^{-1}$, we are in an anion limited regime. The C/A$_{ratio}$ has been calculated for columns, partial columns (up to 1 km) and for the surface, and results for various scenarios are displayed in Figure 10. It should be noted that the ratio has an important month-to-month variability (normalized standard deviation of 10 %) as displayed in Figure 10d for the East Asia region. Note that $NH_{4(p)}^{+}$ is the only cation considered here, the potential role of other cations is discussed below. Figure 10d presents C/A$_{ratio}$



**Figure 10.** a) Partial column (0-1 km) Cations/Anions ratio ($mol_C.mol_A^{-1}$) over China, for the simulations a) 2015A, b) 2015B, c) 2015C, d) monthly variation of C/A$_{ratio}$ over East China. Full lines represent ratios derived from columns (up to 12 km), dashed lines represent ratios derived from 0-1 km column, and dot lines ratios derived from surface concentrations. Black rectangles represent central China and red rectangles Northern China.

for the simulations considering several altitudes, at the surface, the 0-1 km column and the CHIMERE total column. C/A$_{ratio}$ is highly variable depending on considered vertical layer, with significantly lower C/A$_{ratio}$ considering a total column than a reduced layer close to surface, a statement also observable in Paulot et al. (2016). Close to sources (surface) more ammonium will be present, leading often in 2015B and 2015C scenario to a cation exceeding regime. If we consider vertical columns, as 0-1 km for example (which corresponds roughly to the atmospheric mixing layer), the ratio is for most of the months below 1 (anions limited regime). Considering the entire tropospheric column, a cation limited regime occurs for all months. This decrease in the C/A$_{ratio}$ with altitude can be explained by the fact that sulfuric and nitric acid need some time to be formed from precursor gases, while the major cation $NH_4^+$ is directly at surface. Besides, a slight increase of C/A$_{ratio}$ is observed from



January to May, when a maximum is reached. It is probably due to the increase of $NH_3$ emissions during this period compared to decreasing $SO_2$ and $NO_X$ emissions (Figure S1 in supplement file). Then, for July and August, the cations to anions ratio drops. This change is not explained by a change in emissions, because these months display emissions close to June ones. A probable explanation is the following: first July and August correspond to the monsoon season, with higher water vapour

content and solar radiation over the study area, which allow both more OH radical (July and August OH levels are the highest and and twice the annual mean) to form (from $O_3$ photolysis) and second, more $SO_{2(g)}$ dissolution in aqueous phase. Both factors, then induce more $SO_{4(p)}^{2-}$ formation (Stockwell and Calvert, 2016) decreasing by this way the C/A$_{ratio}$.

C/A$_{ratio}$ two-dimensional distributions are shown in Figure 10 for the 0 - 1 km column. The atmosphere is mainly cation limited over East China in the 2015A initial scenario. As expected from the decrease in anion precursor emissions (i.e $SO_2$ and

NOx), the C/A$_{ratio}$ is higher with the 2015C and 2015B simulations than in 2015A simulation, as for example in the Sichuan province-Chongqing municipality (Black Square on Figure 10b) and North China regions (Red rectangle on Figure 10c). Reductions of $SO_2$ and $NO_X$ emissions led to C/A$_{ratio}$ increase and change in the limitation regime close to $NH_{3(g)}$ sources areas (Figure 10a and 10.In the future, emissions reductions for $NH_3$ and anions precursors should lead to less $NH_4NO_{3(p)}$ and $(NH_4)_2SO_{4(p)}$ formation, reducing observed PM levels.

It should also be noted that cations such as $Ca^{2+}$ or $Mg^{2+}$ (from dust or anthropogenic emissions) are not included in CHIMERE chemistry. They could induce a bias in our analysis, underestimating the C/A$_{ratio}$. Nevertheless, we can reasonably assume here that the $NH_{4(p)}^+$ molar content is generally much higher than $Ca^{2+}$ and $Mg^{2+}$ content. A study by Li et al. (2013) with measurements in Beijing, (from September 2006 to August 2007) shows that in winter, when lowest $NH_{4(p)}^+$ concentrations are met, the $NH_{4(p)}^+$ content (about $0.5\,\mu mol.m^{-3}$) still exceeds three times the $(Ca^{2+} + Mg^{2+})$ amount (about

$0.15\,\mu mol.m^{-3}$). Another recent study measuring soluble ions of $PM_{2.5}$ in Beijing shows a large excess, of $NH_{4(p)}^+$ compared to $Ca^{2+}$ or $Mg^{2+}$ in summer and winter 2014 (Chen et al., 2017). Still, not taking into account this chemistry for mineral cations species can lead to simulate cations-limited situations instead of a cations-excess situation for restricted timse of the year and areas, when ratio reaches values close to 1.

### 3.3  Time evolution of inorganic PM and precursor species between 2011 and 2015

Combined impacts of meteorology and emission reductions on gaseous and particulate species are shown in Figure 11 using simulations 2011A, 2013C and 2015C including meteorology and updated emissions for the three corresponding years. The time evolution between 2011 and 2015 is qualitatively similar to that presented in the previous section for emission changes alone. The impact of changing meteorology is to damp the negative changes of pSNA (Figure S11 in supplement file presents two-dimensionnal distribution of pSNA changes) and the positive changes in $NH_{3(g)}$ due to emission reductions. As a result in

our simulations, $NH_{3(g)}$ columns increased by as much as +14 % in 2013 and by 41 % in 2015 over East China, as compared to 2011, combining both meteorological and emission changes.





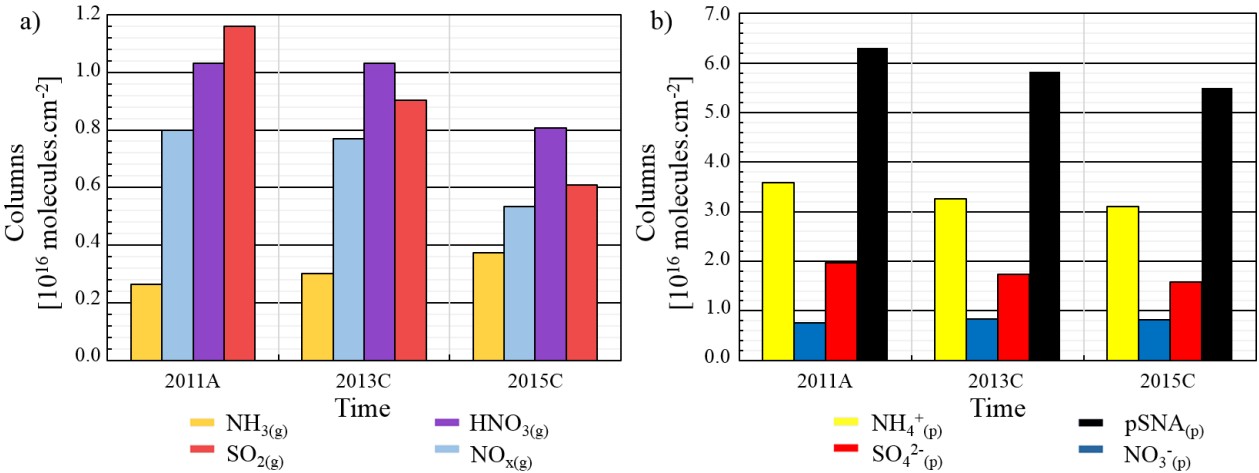

**Figure 11.** a) Evolution of $NO_{X(g)}$, $SO_{2(g)}$, $NH_{3(g)}$ and $HNO_{3(g)}$ tropospheric columns in molecules.$cm^{-2}$, for 2011A, 2013C and 2015C over East China domain, b) evolution of $NH_{4(p)}^{+}$, $SO_{4(p)}^{2-}$, $NO_{3(p)}^{-}$ and pSNA tropospheric columns in molecules.$cm^{-2}$ from 2011 to 2015 over East China domain.

### 3.4 Evaluation against $PM_{2.5}$ surface measurements

The evolution of surface $PM_{2.5}$, induced by changes in $SO_2$ and $NO_X$ emissions in our simulations, is evaluated here against independent daily $PM_{2.5}$ surface measurements. Scores for normalized bias, normalized RMSE, ratios of model and observed variability, and Pearson correlation coefficients are displayed in Table 3. We used data from U.S. Embassy and consulates over

China (e.g., in Beijing, Chengdu, Guangzhou, Shanghai and Shenyang, www.stateair.net, last consulted 10/05/2018), for the years 2013 and 2015. For the year 2011, data are available only for Beijing, so this year was discarded. We present results for updated inventories (simulation 2013C and 2015C). We also present changes, between 2013A/2015A and 2013C/2015C simulations in Table 3.

     The $PM_{2.5}$ values measured in Chinese cities show large amplitudes, ranging from 0 to $500\,\mu g.m^{-3}$, and CHIMERE cor-

rectly represents both these amplitudes and the strong day-to-day variability of $PM_{2.5}$ surface concentrations, as illustrated by good correlation coefficients (Pearson coefficient averaging 0.7) between time series and a ratio of standard deviations close to unity (except for Shenyang, table 3). Our 2013C and 2015C simulations overestimate average $PM_{2.5}$ concentrations for 4 cities (Beijing, Shanghai, Guangzhou and Chengdu), and underestimate them for Shenyang (Bias=-13 %). For the Beijing, Shanghai, Guangzhou and Chengdu stations, we observe a slight decrease of $PM_{2.5}$ means ($-4.3\,\mu g.m^{-3}$; $-2.2\,\mu g.m^{-3}$;

$-1.1\,\mu g.m^{-3}$; $-11.5\,\mu g.m^{-3}$) between our reference simulations (2013A and 2015A) and simulations with modified inventories (2013C and 2015C). This means that updating the emissions improves agreement with observations by reducing biases and errors (NRMSE). We also find a slight decrease at Shenyang station ($-1.1\,\mu g.m^{-3}$), slightly deteriorate the already negative bias. The strongest improvement is observed in Chengdu (central China), an area where inorganic PM mainly depends on





sulphate and ammonium. $PM_{2.5}$ observations show, between 2013 and 2015, a decrease for Beijing, Chengdu, Guangzhou

**Table 3.** Daily $PM_{2.5}$ comparison between model and measurements for 2013C and 2015C. "Changes" corresponds to differences between 2013C and 2015C comparisons on one hand and 2013A and 2015A ones on the other. Bias and NRMSE are normalized using the measurement mean. R corresponds to the Pearson correlation coefficient and n represents the number of available daily means.

| Stations | $PM_{2.5}$ Measurement mean ($\mu g.m^{-3}$) | $PM_{2.5}$ Model mean($\mu g.m^{-3}$) / Changes ($\mu g.m^{-3}$) | Bias(%) / Changes(%) | NRMSE(%) / Changes(%) | $\dfrac{\sigma_{CHIMERE}}{\sigma_{obs}}$ | R / Changes | n |
|---|---|---|---|---|---|---|---|
| **Beijing** | 92.3 | 115.7 / -4.3 | +25 % / -05 % | 64 % / -03 % | 1.0 | 0.77 / = | 730 |
| **Shanghai** | 55.3 | 68.6 / -2.2 | +24 % / -04 % | 57 % / -04 % | 1.1 | 0.76 / = | 723 |
| **Guangzhou** | 47.6 | 59.8 / -1.1 | +26 % / -02 % | 64 % / -01 % | 1.0 | 0.54 / = | 719 |
| **Chengdu** | 83.8 | 127.6 / -11.5 | +52 % / -14 % | 71 % / -12 % | 1.0 | 0.72 / +0.03 | 687 |
| **Shenyang** | 74.0 | 64.4 / -1.0 | +13 % / -02 % | 61 % / -01 % | 0.6 | 0.68 / +0.02 | 570 |

and Shanghai respectively of -19 %, -20 %, -30 % and -15 % and an increase for Shenyang of +18 %. These changes are not fully reproduced by CHIMERE, possibly, as emissions have been modified for $NO_X$ and $SO_2$ only, not considering organic or other inorganic species. In CHIMERE, $PM_{2.5}$ decreases are calculated for Beijing and Chengdu (respectively of -3.6 % and

-10 %), no significant change is simulated for Guangzhou and increases are modelled for Shanghai and Shenyang (respectively of +12.5 % and +3.7 %). All changes are calculated filtering simulation results according to measurements availability. Increase in Shenyang can be explained due to lacking data sampling periods between 2013 and 2015 (229 days available against 341). In Shanghai, the modelled increase is not explained by $PM_{2.5}$ inorganic components, which present a +1 % trend at surface, but due to meteorological condition effects on $PM_{2.5}$ others components present larger increases.

**3.5  Evaluation against IASI $NH_{3(g)}$ columns observations**

Data retrieved from IASI instrument allow us to compare satellite observations to simulations and to verify the consistency of simulated trends. Figure 12 shows the spatial distributions of ammonia over China for IASI and CHIMERE for 2011, 2013 and 2015, presenting a similar spatial pattern and a good correlation ($\sim R_{IASI-CHIMERE} = 0.91$ over East China) and an acceptable daily correlation ($\sim R_{IASI-CHIMERE} = 0.55$). Nevertheless, simulations underestimate ammonia levels with a

bias of -39 %, mainly because of comparison over sea area. As described above, IASI observations show a +65 % increase between 2011 and 2015. Interestingly, our model results between 2011A and 2015C, thus taking into account emission and meteorology changes, show a rather similar difference of +49 % of $NH_{3(g)}$ (when CHIMERE simulations are sampled on daily IASI observations availability). For the intermediate year 2013, the IASI satellite observed a +15 % increase in $NH_{3(g)}$ columns, while CHIMERE simulations showed a +24 % increase for 2013C scenario (again sampled on IASI observations).

On the contrary, simulations with unmodified emissions only show small changes for both years (+6 % in 2013A; -3 % in



2015A). (Liu et al., 2018, in discussion) estimated a +33 % NH$_3$ columns increase over the North China Plain, between 2011 and 2015, taking account of SO$_2$ emissions decreases, a value close to our result for this case (+27 % between 2011A and 2015B).This suggests that the observed increase for ammonia by IASI can be, to a large part, explained by changes in atmospheric chemistry induced by SO$_2$ but also by NO$_X$ emissions reductions, with less ammonium present within inorganic aerosol and more ammonia remaining in the gas phase.

**Figure 12.** Ammonia columns evolution for IASI (top) and CHIMERE (bottom) a) 2011, b) 2013, c) 2015., d) 2011A, e) 2013C and f) 2015C in molecules.cm$^{-2}$.

## 4   Conclusion

Sensitivity tests with the regional chemistry-transport model CHIMERE have been performed to understand the evolution of the NH$_{3(g)}$ atmospheric content over China, with an increase observed by IASI measurements over Eastern China of +15 %



between 2011 and 2013, and of +65 % between 2011 and 2015. One of the main results of this study is that the strong observed changes in the $NH_{3(g)}$ atmospheric content are mainly associated with a reduction of anthropogenic $SO_2$ emissions, and to a lesser extent, to a reduction of anthropogenic $NO_X$ emissions and to interannual changes in meteorological conditions. Indeed, with $SO_2$ emissions reduced by 24 % between 2011 and 2013 and by -37.5 % between 2011 and 2015, with an additional $NO_X$

emissions reduction of -21 % between 2011 and 2015, CHIMERE reproduces an increase of $NH_{3(g)}$ atmospheric content of +24 % between 2011 and 2013 and of +49 % between 2011 and 2015 (when filtering simulations according to IASI observations availability). Also, it should be recalled that $NH_3$ emissions have remained constant in our scenarios, as no precise information on $NH_3$ emission changes was available to us but Zhang et al. (2018) suggested a 7 % increase between 2011 and 2015 what could partly explain the difference between IASI and CHIMERE increases. $SO_2$ and $NO_X$ emission reductions have been

inferred from OMI satellite observations, so our study is to a large degree constrained by observations. Simulations allow then to state that observed $SO_2$ and $NO_X$ columns decreases and $NH_3$ increases are mutually consistent.

The cation to anion ratio shows an interesting height dependence. It is below unity for total columns, above unity for surface, and near unity for the first km of the atmosphere. The latter is probably most relevant for inorganic aerosol formation affecting air quality. Thus it appears that in addition to $SO_2$ and $NO_X$ reductions, also $NH_3$ emission reductions would be

efficient to reduce inorganic aerosol formation. The reduction of $SO_2$ and $NO_X$ emissions also leads to a decrease of inorganic pSNA production (-14 % in the tropospheric columns between 2011 and 2015) which is highly contributing to the $PM_{2.5}$ concentrations (about 50 % of surface $PM_{2.5}$, and 33 % of column PM). A shift from sulphate to nitrate is simulated, due to stronger $SO_2$ than $NO_X$ reductions, and also more ammonia available for nitrate formation.

Finally, from our work, it appears that the evolution of gaseous precursors must be updated each year to understand the PM

evolution. Current bottom-up inventories are not up-dated quickly enough. The method we used to derived inventories for 2013 and 2015 from satellite data provides an interesting first estimation but presents uncertainties when pollutants are transported or eliminated. Consequently, it would be interesting to use inverse methods operating in synergy between regional CTM and atmospheric observations (i.e, DECSO; Mijling and Zhang, 2013) to better represent $NO_X$ and $SO_2$ emissions. Inverse modelling systems could also be used to quantify $NH_3$ emissions, as IASI space-based $NH_3$ observations indeed proved its

considerable potential to inform about the high spatio-temporal variability of $NH_3$ emissions (Fortems-Cheiney et al., 2016).

**Authors contributions**

M.L. and A.F.-C. designed the experiments and M.L. carried them out. L.C., C.C., P.-F.C. and M.V.D. were responsible for the satellite retrieval algorithm development and the processing of the IASI $NH_3$ dataset. G.S. prepared meteorological and emission data. A.F.-C. prepared emission update and satellite data. M.L. adapted the model code and performed the simulations.

M.L. prepared the manuscript and all authors contributed to the text, interpretation of the results and reviewed the manuscript.



*Acknowledgements.* We acknowledge the free use of tropospheric $NO_2$ column data from the OMI sensor from www.temis.nl. The thesis of M. Lachatre was funded by Sorbonne Universités and this study was funded by PolEASIA ANR project under the allocation ANR-15-CE04-0005. This work was granted access to the HPC resources of TGCC under the allocation A0030107232 made by GENCI. $PM_{2.5}$ measurements provided by U.S. Department of State Air Quality Monitoring Program, Mission China. IASI is a joint mission of Eumetsat and

5    the Centre National d'Études Spatiales (CNES, France). The authors acknowledge the Aeris data infrastructure (https://www.aeris-data.fr/) for providing access to the IASI Level-2 $NH_3$ data used in this study. The French scientists are grateful to CNES and Centre National de la Recherche Scientifique (CNRS) for financial support. The research in Belgium is also funded by the Belgian State Federal Office for Scientific, Technical and Cultural Affairs and the European Space Agency (ESA Prodex IASI Flow project).



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
