# Peer review of "The unintended consequence of $SO_2$ and $NO_2$ regulations over China: increase of ammonia levels and impact on $PM_{2.5}$ concentrations"

_Atmospheric Chemistry and Physics, 2018_

## Referee Comment (RC1) · Anonymous Referee #1 · 14 Dec 2018

This paper investigated the reason for the increase of atmospheric concentrations of NH3 in China. The authors compared model simulations for 2011, 2013 and 2015 to examine inter-annual change of meteorology and reductions in SO2 and NOX emission. The results are useful for PM pollution control in China. Similar topic and conclusions have been shown in at least two recent studies studies (Fu et al., 2017, Liu et al., 2018). It is important to highlight the difference and new insights in the present work. In addition, the paper requires extensive English editing.

Specific comments:

1. Page 4, Line 6: Meteorology predictions need to be validated before exploring its

impacts on NH3 concentrations.

2. It's better to put the model validation part (section 3.4 and 3.5) to the first part of section 3, because it's the foundation of the following analysis. Validation of SO2 and NOX predictions need to be added.

3. Page 3, Line 6-7: Why the operationally provided IASI level 2 data cannot be used to analyze the inter-annual NH3 variability?

4. Page 4: In the EDGAR-HAP-v2.2 inventory you used for 2010, Chinese emissions are derived from the MEIC inventory. The MEIC inventories for 2012, 2014 and 2016 are available in its website (http://www.meicmodel.org/). Why not use the MEIC inventory directly for 2013 and 2015? What is difference between your updated emissions for 2013 and 2015 and those in MEIC?

Minor comments:

Page 1, line 4: The full name for "IASI" need to be given.

Page 2, line 5: "NH3(g) Chinese emissions " should be "NH3(g) emissions in China"

Page 2, line 23: "observed" should be deleted

Page 2, line 25: "ran" should be "conducted"

Page 10, line 2: "reaction" should be deleted.

Page 12, line 16 to Page 13, line 2: The English grammar for the last sentence need to be checked.

Page 16, line 4-7: It's difficult to understand these sentences, and the statement need to be improved.

Page 18: It's difficult to read Table 3. Better presentation and interpretation are needed.

References

Fu, X., Wang, S., Xing, J., Zhang, X., Wang, T., and Hao, J.: Increasing Ammonia Concentrations Reduce the Effectiveness of Particle Pollution Control Achieved via SO2 and NOX Emissions Reduction in East China, Environmental Science and Technology Letters, 4, 221–227.

Liu, M., Huang, X., Song, Y., Xu, T., Wang, S., Wu, Z., Hu, M., Zhang, L., Zhang, Q., Pan, Y., and Zhu, T.: Rapid SO2 emission reductions significantly increase tropospheric ammonia concentrations over the North China Plain, Atmospheric Chemistry and Physics Discussions, 2018.

---

## Referee Comment (RC2) · Anonymous Referee #2 · 23 Jan 2019

The manuscript studies the changes in atmospheric concentration of ammonia over China for the recent past. The authors constructed emissions and performed model simulations with a regional model to investigate thee reasons behind the change in concentrations. The results are compared to satellite data.

The manuscript is in general well written, but there is a need for English language editing.

The findings are useful for understanding the PM concentrations over China and potentially managing PM pollution . Specific comments:

P1L5: add the % sign to -37.5

[Figure]

P1L6: the (g) in $NO_3$ is redundant as gaseous is mentioned

P2L3: This abbreviation has not been defined yet

P2L5: "Chinese emissions" can be removed since it is mentioned later in the sentence that you are

talking about Chinese emissions P2L7: ...in 2005 and *have* been...

P2L13: Is likewise the proper word to use here?

P2L13-15: Maybe break the sentence in two. The way it is now it is not easy to understand.

P2L15: put $NH_4NO_3$ and $HNO_3$ in parentheses.

P2L21: Use proper citation formatting for reference (Liu et al)

P3L6: change the quote style in "climate". Use the same quotation style throughout the document

P3L6: ECMWF abbreviation has not been defined yet

P4L15: Since you are creating emissions for years 2013, 2015, why not create emissions for 2011 also? Isn't this adding to your uncertainty?

P5L3: is piloted the correct word to use here?

P5L7: This is confusing. Basically you apply a factor that is 2011 based, on an emission inventory of 2010. How does this affect your calculations? You should either make it all 2010 based, or all 2011 based.

P7L24: correct typo on EDHAR to EDGAR

P8L3: Here you report more than 90 %, but later more than 95%. You should be more consistent.

P10L2: A comma is needed after R4

P10L12-14: either...or, not or....either

P11L13: As before, be consistent on the numbers you report.

P16L22: correct the typo on times

P16L28: All your scenarios use the respective year's meteorology. How do you attribute differences caused by emissions changes to meteorology, since everything in the model changes, except for the NOx and SOx emissions?

P18 table: remove the zeros from the beginning of non decimal numbers

P19L1: Change the citation style on Liu et al

P20L4: Either report both reductions as negative numbers, or both as positive numbers.

P20L20: updated, not up-dated

---

## Author Comment (AC1) · 3 Apr 2019

We wish to thank the referee for his/her helpful comments. The comments of the referee are in bold and our answers in normal black.

Similar topic and conclusions have been shown in at least two recent studies (Fu et al., 2017, Liu et al., 2018). It is important to highlight the difference and new insights in the present work.

Fu et al., (2017), Liu et al., (2018) were already mentioned in manuscript initial version and more details about new insights from our study have been added during revision.

[Figure]

The present study brings new insights concerning NOx emissions reduction impact on ammonia, described P12L9. In the study Liu et al., (2018) do not process to a SO2 emissions changes only simulation, which has shown in our study a large increase of nitrate production and helped us to figure out that change of SO2 and NOx emissions combine have produce more NH3 released in the gas phase than SO2 emissions changes alone. Fu et al., 2017 conclusion are considered and compared P15L29-L30. If our results agree with those presented in Fu et al., 2017, it brings a more precise view of Inorganic PM system with the insight brought by the cation / anion ratio and altitude analysis. This work on PM helped us to understand nitrates conservation (mentioned in Liu et al., 2018 from ground measures) between 2011 and 2015. Also, we have used information from IASI instrument to evaluate modelled NH3 evolution.

We added a sentence P3L5: "A very recent study by Liu et al. (2018) suggests that ammonia increase mainly comes from SO2 emission policies. They found that the changes in NOX emissions decreased the NH3 column concentrations in their study period. On the contrary, Fu et al. (2017) have shown that SO2 and NO2 emissions control was an important factor affecting the significant enhancement of NH3 column concentrations over China during the period 2011–2014. In addition, our study also presents a comparison to NH3 IASI satellite observations."

We added a sentence P12L9: "This statement on NOX emission evolution impacts is different from that in Liu et al. (2018), in which NOX emission reduction is considered as not responsible for the NH3 increase between 2011 and 2015." This additional NOx emission dependence is an important and original point of our study.

We modified a sentence P19L12 "Liu et al. (2018) estimated a +35% NH3 columns increase over the North China Plain, between 2011 and 2015, taking account of SO2 emissions decrease, a value close to our result for this case (+27% between 2011A and 2015B)."

We added a sentence P15L29-L30 "In the future, emissions reductions for NH3 and

anions precursors should lead to less NH4NO3(p) and (NH4)2SO4(p) formation, reducing observed PM levels, which was already suggested in Fu et al. (2017)."

In addition, the paper requires extensive English editing. English editing has been performed with the help of a native English speaking colleague.

Specific comments:

1. Page 4, Line 6: Meteorology predictions need to be validated before exploring its impacts on NH3 concentrations. In this case we use a meteorological fields provided by the Integrated Forecasting System of ECMWF which is an operational product extensively validated by the center (Owens and Hewson, 2018). As an example, ECMWF Zenith Tropospheric Delay (ZTD) has been evaluated from GPS ZTD (Chen et al., 2010) the bias ranged from 11.5 to -28.6 mm with a corresponding average of -10.5 mm. Jingjing et al., 2015 evaluated Planetary Boundary Layer Height with CALYPSO. Moreover this product is based on meteorological analysis, which means that observations (in situ, satellite) are used to correct the initial state of the model every 6 hours which is, for temperature, humidity, a guarantee of the good quality of the fields.

2. It's better to put the model validation part (section 3.4 and 3.5) to the first part of section 3, because it's the foundation of the following analysis. In our study, we assume that our main result on ammonia increase (section 3.5) should be kept as the final part of our paper, just before the conclusion, as the IASI/CHIMERE comparison and evaluation. It is the final point of our paper, which validate the consistency of hypothesis made on emissions and meteorological changes, investigated separately in section 3.1 and 3.2.

Validation of SO2 and NOX predictions need to be added. SO2 and NOx columns predictions from emissions update have indeed been compared to the OMI satellite evolution in Part 2.2. It should be recalled that our emission estimations for SO2 and NOx have also been compared and are consistent with the new MEIC inventory. (See below, answer to 4.)
We added a sentence (P7L5): ""Emission update allowed to reproduce correctly SO2 and NO2 column evolutions, with for SO2 -44% (CHIMERE) and -53% (OMI) between 2011 and 2015, and for NO2 -31% (CHIMERE) and -23% (OMI) between 2013 and 2015."

3. Page 3, Line 6-7: Why the operationally provided IASI level 2 data cannot be used to analyze the inter-annual NH3 variability? This is fully explain in Van Damme et al., 2017 : "The analysis of ANNI-NH3-v2.1 time series revealed several sharp discontinuities which seemed to coincide with IASI L2 version changes (see Fig. 3). In particular, a noticeable overall increase in the NH3 columns was found to correspond with the change from v5 to v6, and a smaller decrease was observed with the introduction of v6.2. As we will show below, these are a direct consequence of algorithmic changes to the retrieved temperature of the surface and lower troposphere. Following these findings, the need arose for a self-consistent IASI NH3 dataset, which uses stable and uniform input data. The ECMWF ERA-Interim reanalysis (Dee et al., 2011) is very suitable for this purpose, as it provides all the necessary meteorological parameters and covers the whole IASI time period."

We added the following sentences in the text (P4L7): "For this study we used the dataset ANNI-NH3-v2.2R-I, relying on ERA-Interim ECMWF (European Centre for Medium-Range Weather Forecasts) meteorological input data rather than the operationally provided Eumetsat IASI Level 2 (L2) data used for the standard near-real-time version. The analysis of ANNI-NH3-v2.1 time series indeed revealed sharp discontinuities coinciding with IASI L2 version changes (Van Damme et al., 2017). With the ECMWF ERA-Interim reanalysis, the time series is now coherent in time (excepted for the cloud coverage flag) and can therefore be used to study interannual NH3(g) variability over East China between 2011 and 2015 (Figure 1)."

4. Page 4: In the EDGAR-HTAP-v2.2 inventory you used for 2010, Chinese emissions are derived from the MEIC inventory. The MEIC inventories for 2012, 2014 and 2016 are available in its website (http://www.meicmodel.org/). Why not use the MEIC inventory directly for 2013 and 2015? What is difference between your updated emissions for 2013 and 2015 and those in MEIC? We have initiated our work on ammonia since mid 2016, and at this time, updated emissions inventories were not available..

We now compare latest MEIC inventory (Zheng et al., 2018, Figure 1) to our updated emissions, which brings similar evolution of the emissions between 2011 and 2015. We added the following sentences: P6L12: "A recent study from Zheng et al. (2018) evaluated NOX emissions evolution of -17.4% between 2011 and 2015, similar to our -24% evolution."

P7L4: "Zheng et al. (2018) evaluated SO2 emissions evolution of -41.9% between 2011 and 2015, again similar to our -37.5% evolution."

Minor comments:

Page 1, line 4: The full name for "IASI" need to be given. Included Page 2, line 5: "NH3(g) Chinese emissions " should be "NH3(g) emissions in China" Modified

Page 2, line 23: "observed" should be deleted Modified

Page 2, line 25: "ran" should be "conducted" Modified

Page 10, line 2: "reaction" should be deleted. Modified

Page 12, line 16 to Page 13, line 2: The English grammar for the last sentence need to be checked. Sentence has been reformulated P13L9:

"However, for conditions of weak atmospheric dispersion or high humidity, as in the Sichuan province and the Chongqing municipality (located in an orographic depression), sulphates can be formed formed closer to sources. In this area, sulphates largely contribute to the PM column, as much as 32% as compared to 23% over East China, SO2-4(p), see Figure S8 in supplement file"

Page 16, line 4-7: It's difficult to understand these sentences, and the statement need to be improved. Sentence has been reformulated P15L20: "A probable explanation is

the following: first July and August correspond to the monsoon season, with higher water vapour content and solar radiation over the study area. This leads to enhanced OH radical concentrations (up to twice the annual mean) to form $H_2SO_4(g)$ and $HNO_3(g)$. Second, higher water content inducts more $SO_2(g)$ dissolution in aqueous phase. Both factors, then induce more $SO_4^{2-}(p)$ formation (Stockwell and Calvert,2016) decreasing by this way the C/A ratio."

Page 18: It's difficult to read Table 3. Better presentation and interpretation are needed. Table has remained identical but we tried to be more explicit in Table cation to help reader to quickly understand what "Changes" are indicating in Table 3 P18. "Table 3. Daily PM2.5 comparison between model and measurements for 2013C and 2015C simulations. "Changes" corresponds to differences between 2013C and 2015C comparisons on one hand and 2013A and 2015A ones on the other (i.e. BiasChanges = BiasC - BiasA). Bias and NRMSE are normalized using the measurement mean. R corresponds to the Pearson correlation coefficient and n represents the number of available daily means."

References:

Chen, Q., Song, S., Stefan, H., Yuei-An, L., Zhu, W., and Jingyang, Z.: Assessment of ZTD derived from ECMWF/NCEP datawith GPS ZTD over China, GPS Solut, https://doi.org/10.1007/s10291-010-020, 2010. Fu, X., Wang, S., Xing, J., Zhang, X., Wang, T., and Hao, J.: Increasing Ammonia Concentrations Reduce the Effectiveness of Particle Pollution Control Achieved via SO2 and NOX Emissions Reduction in East China, Environmental Science and Technology Letters, 4, 221–227, https://doi.org/10.1021/acs.estlett.7b00143, https://doi.org/10.1021/acs.estlett.7b00143, 2017. Jingjing, L., Jianping, H., Bin, C., Tian, Z., Hongru, Y., Hongchun, J., Zhongwei, H., and Beidou, Z.: Comparisons of PBL heights derived from CALIPSO and ECMWF reanalysis data over China, Journal of Quantitative Spectroscopy and Radiative Transfer, 153, 102 – 112,https://doi.org/https://doi.org/10.1016/j.jqsrt.2014.10.011,http://www.sciencedirect.com/science/article/pii/S002240731

topical issue on optical particle characterization and remote sensing of the atmosphere: Part II, 2015. Liu, M., Huang, X., Song, Y., Xu, T., Wang, S., Wu, Z., Hu, M., Zhang, L., Zhang, Q., Pan, Y., and Zhu, T.: Rapid SO2 emission reductions significantly increase tropospheric ammonia concentrations over the North China Plain, Atmospheric Chemistry and Physics Discussions, 2018, 1–19, https://doi.org/10.5194/acp-2018-880, https://www.atmos-chem-phys-discuss.net/acp-2018-880/, 2018. Owens, R. G. and Hewson, T.: ECMWF Forecast User Guide, Reading, https://doi.org/10.21957/m1cs7h, https://software.ecmwf.int/wiki/display/FUG/Forecast+User+Guide,  Replaces previous editions that were available as PDF documents., 2018. Van Damme, M., Whitburn, S., Clarisse, L., Clerbaux, C., Hurtmans, D., and Coheur, P.-F.: Version 2 of the IASI NH3 neural network retrieval algorithm: near-real-time and reanalysed datasets, Atmos. Meas. Tech., 10, 4905-4914, https://doi.org/10.5194/amt-10-4905-2017, 2017. Zheng, B., Tong, D., Li, M., Liu, F., Hong, C., Geng, G., Li, H., Li, X., Peng, L., Qi, J., Yan, L., Zhang, Y., Zhao, H., Zheng, Y., He, K., and 35 Zhang, Q.: Trends in China's anthropogenic emissions since 2010 as the consequence of clean air actions, Atmospheric Chemistry and Physics, 18, 14 095–14 111, https://doi.org/10.5194/acp-18-14095-2018, https://www.atmos-chem-phys.net/18/14095/2018/, 2018.

[Figure]

**Fig. 1.** Emissions evolution in China, from 2010 to 2017 (in Tg.yr-1) from Zheng et al, 2018

---

## Author Comment (AC2) · 3 Apr 2019

We wish to thank the referee for his/her helpful comments. The comments of the referee are in bold and our answers in normal black.

The manuscript is in general well written, but there is a need for English language editing.

English language editing has been performed with help of a native English speaker.

[Figure]

Specific comments:

P1L5: add the % sign to -37.5
Modified

P1L6: the (g) in NO3 is redundant as gaseous is mentioned
In some part of the document species is only indicated with the indexes (p) or (g), I'll conserve them for all cases to preserve document homogeneity.

P2L3: This abbreviation has not been defined yet
Modified

P2L5: "Chinese emissions" can be removed since it is mentioned later in the sentence that you are talking about Chinese emissions P2L7: ...in 2005 and have been.
Removed

P2L13: Is likewise the proper word to use here? And P2L13-15: Maybe break the sentence in two. The way it is now it is not easy to understand. And P2L15: put NH4NO3 and HNO3 in parentheses
Modified

P2L12:
"The rise of ammonia concentrations over China could be explained by increased NH3(g) evaporated from inorganic PM due to a rise in temperature. As shown by Riddick et al. (2016), meteorological variations would change both the NH3(g)

volatilization and the equilibrium between ammonia, ammonium nitrate (NH4NO3) and nitric acid (HNO3)."

P2L21: Use proper citation formatting for reference (Liu et al)
Modified

P3L6: change the quote style in "climate". Use the same quotation style throughout the document
Modified

P3L6: ECMWF abbreviation has not been defined yet
Added

P4L15: Since you are creating emissions for years 2013, 2015, why not create emissions for 2011 also? Isn't this adding to your uncertainty?
This could have been an option; we did not proceed to this modification as it appeared that 2010/ 2011 (+4.5Also, please keep in mind that this procedure only affects the absolute values of emissions, and by much less than the currently accepted uncertainty for emissions of several tenths of percent. The relative changes between 2011, 2013 and 2015 are correctly taken into account (see our answer just below (P5L7).

P5L3: is piloted the correct word to use here?
Modified by "controlled"

P5L7: This is confusing. Basically you apply a factor that is 2011 based, on an

emission inventory of 2010. How does this affect your calculations? You should either make it all 2010 based, or all 2011 based
As said above (P4L15), our procedure slightly affects the absolute level of emissions (a few percents), but the relative changes between 2011 and 2015 are correctly taken into account. In addition, taking the year 2010 emissions which was the last year in the EDGAR inventory that we used, allowed us using uncorrected emissions. Indeed, correcting these emissions with our satellite columns based method also induces uncertainty. At the end, we use 2010 emissions to simulate 2011 and we then update emissions based on 2011-2015 satellite observation changes to evaluate change on ammonia between 2011 and 2015.

P7L24: correct typo on EDHAR to EDGAR
Modified

P8L3: Here you report more than 90 Modified

P10L2: A comma is needed after R4
Added

P10L12-14: either...or, not or....either
Modified

P16L22: correct the typo on times
Modified

P16L28: All your scenarios use the respective year's meteorology. How do you

at-tribute differences caused by emissions changes to meteorology, since everything in the model changes, except for the NOx and SOx emissions?

In this part, we combine information and conclusion from sections 3.1 (Impact of meteorological conditions on the ammonia/ammonium/sulphate/nitrate system; Fig. 5) and 3.2 (Impact of SO2 and NOX emission reduction on NH3 columns and inorganic aerosol; Fig 9) to better understand changes in section 3.3, which give the response both the whole of meteorology and emissions changes.

Section and figure have been added P17L5: "The impact of changing meteorology is to damp the negative changes of pSNA (Section 3.1, Figure 5 and Figure S11 in supplement file presents two-dimensionnal distribution of pSNA changes) and the positive changes in NH3(g) due to emission reductions."

P18 table: remove the zeros from the beginning of non decimal numbers
I'd prefer not to change this because it keeps the table better structured and more easily readable.

P19L1: Change the citation style on Liu et al.
Modified

P20L4: Either report both reductions as negative numbers, or both as positive numbers
Modified

P20L20: updated, not up-dated

Modified

References:

Chen, Q., Song, S., Stefan, H., Yuei-An, L., Zhu, W., and Jingyang, Z.: Assessment of ZTD derived from ECMWF/NCEP datawith GPS ZTD over China, GPS Solut, https://doi.org/10.1007/s10291-010-020, 2010. Fu, X., Wang, S., Xing, J., Zhang, X., Wang, T., and Hao, J.: Increasing Ammonia Concentrations Reduce the Effectiveness of Particle Pollution Control Achieved via SO2 and NOX Emissions Reduction in East China, Environmental Science and Technology Letters, 4, 221–227, https://doi.org/10.1021/acs.estlett.7b00143, https://doi.org/10.1021/acs.estlett.7b00143, 2017.

Jingjing, L., Jianping, H., Bin, C., Tian, Z., Hongru, Y., Hongchun, J., Zhongwei, H., and Beidou, Z.: Comparisons of PBL heights derived from CALIPSO and ECMWF reanalysis data over China, Journal of Quantitative Spectroscopy and Radiative Transfer, 153, 102 – 112,https://doi.org/https://doi.org/10.1016/j.jqsrt.2014.10.011,http://www.sciencedirect.com/science/article/pii/S002240731 topical issue on optical particle characterization and remote sensing of the atmosphere: Part II, 2015.

Liu, M., Huang, X., Song, Y., Xu, T., Wang, S., Wu, Z., Hu, M., Zhang, L., Zhang, Q., Pan, Y., and Zhu, T.: Rapid SO2 emission reductions significantly increase tropospheric ammonia concentrations over the North China Plain, Atmospheric Chemistry and Physics Discussions, 2018, 1–19, https://doi.org/10.5194/acp-2018-880, https://www.atmos-chem-phys-discuss.net/acp-2018-880/, 2018.

Owens, R. G. and Hewson, T.: ECMWF Forecast User Guide, Reading, https://doi.org/10.21957/m1cs7h, https://software.ecmwf.int/wiki/display/FUG/Forecast+User+Guide,  Replaces previous editions that were available as PDF documents., 2018.

Van Damme, M., Whitburn, S., Clarisse, L., Clerbaux, C., Hurtmans, D., and Coheur, P.-F.: Version 2 of the IASI NH3 neural network retrieval algorithm: near-real-time and reanalysed datasets, Atmos. Meas. Tech., 10, 4905-4914, https://doi.org/10.5194/amt-10-4905-2017, 2017.

Zheng, B., Tong, D., Li, M., Liu, F., Hong, C., Geng, G., Li, H., Li, X., Peng, L., Qi, J., Yan, L., Zhang, Y., Zhao, H., Zheng, Y., He, K., and 35 Zhang, Q.: Trends in China's anthropogenic emissions since 2010 as the consequence of clean air actions, Atmospheric Chemistry and Physics, 18, 14 095–14 111, https://doi.org/10.5194/acp-18-14095-2018, https://www.atmos-chem-phys.net/18/14095/2018/, 2018.
* * *
[Figure]

**Fig. 1.** Emissions evolution in China, from 2010 to 2017 (in Tg.yr-1) from Zheng et al, 2018.